# Silver Nanoparticles Biocomposite Films with Antimicrobial Activity: In Vitro and In Vivo Tests

**DOI:** 10.3390/ijms231810671

**Published:** 2022-09-14

**Authors:** Anca Niculina Cadinoiu, Delia Mihaela Rata, Oana Maria Daraba, Daniela Luminita Ichim, Irina Popescu, Carmen Solcan, Gheorghe Solcan

**Affiliations:** 1Faculty of Medical Dentistry, “Apollonia” University of Iasi, 700511 Iasi, Romania; 2Petru Poni Institute of Macromolecular Chemistry, 700487 Iasi, Romania; 3Faculty of Veterinary Medicine, “Ion Ionescu de la Brad” Iasi University of Life Sciences, 700489 Iasi, Romania

**Keywords:** biocomposite films, silver nanoparticles, ibuprofen, *Staphylococcus aureus*

## Abstract

Overuse of antimicrobials by the population has contributed to genetic modifications in bacteria and development of antimicrobial resistance, which is very difficult to combat nowadays. To solve this problem, it is necessary to develop new systems for the administration of antimicrobial active principles. Biocomposite systems containing silver nanoparticles can be a good medical alternative. In this context, the main objective of this study was to obtain a complex system in the form of a biocomposite film with antimicrobial properties based on chitosan, poly (vinyl alcohol) and silver nanoparticles. This new system was characterized from a structural and morphological point of view. The swelling degree, the mechanical properties and the efficiency of loading and release of an anti-inflammatory drug were also evaluated. The obtained biocomposite films are biocompatibles, this having been demonstrated by in vitro tests on HDFa cell lines, and have antimicrobial activity against *S. aureus*. The in vivo tests, carried out on rabbit subjects, highlighted the fact that signs of reduced fibrosis were specific to the C2P4.10.Ag1-IBF film sample, demonstrated by: intense expression of TNFAIP8 factors; as an anti-apoptotic marker, MHCII that favors immune cooperation among local cells; αSMA, which marks the presence of myofibroblasts involved in approaching the interepithelial spaces for epithelialization; and reduced expression of the Cox2 indicator of inflammation, Col I.

## 1. Introduction

A serious problem facing the population of the world is that the use of antibiotics has led to the emergence of multi-resistant microorganisms, which are very difficult to combat [1]. Development of new drug delivery systems that can extend half-life, improve bioavailability, optimize pharmacokinetics, and decrease dosing frequency of drugs could be an excellent solution to these issues [2]. Among various drug delivery systems, biocomposite films can be a promising alternative, because they are made from materials from natural and renewable sources and present multiple advantages, such as nontoxicity and an increased degree of flexibility, one similar to natural tissue [3,4]. Local drug delivery releases the drugs at high concentrations at the site of infection; compared to systemic antibiotic therapy, it provides a more targeted delivery, increased bioavailability, self-administration and reduced side effects [5]. More and more researchers have become interested in the strategy of treating antibiotic-resistant bacterial infections, trying to explore new strategies without antibiotics or based on the administration of lower doses of drugs [6]. Incorporation of nanostructures into hydrogels leads to the development of complex systems which can improve cell adhesion, cell organization, and cell–cell interactions in order to create tissue constructs with enhanced mechanical integrity, electroactivity, and improved cellular organization [7]. Science and technology are working in synergistic combination to produce new materials at the nanoscale level [8], and various metallic and ionic nanomaterials have been widely investigated, including copper, zinc, titanium, magnesium, gold and silver; silver nanoparticles (AgNPs) have shown the highest efficacy against bacteria and viruses and fungi [8,9]. Silver nanoparticles (AgNPs) have drawn significant attention for their various uses in antimicrobial gel formulations, orthopedic applications, catheters and medical instruments, implants and contact lens coatings. Due to the physical, chemical and biological characteristics of AgNPs and their potential to be incorporated into biocomposite materials, especially hydrogel scaffolds, they are also used for burn and wound healing [10]. Likewise, AgNp are alternative antibacterial agents to antibiotics and have the ability to overcome the bacterial resistance against antibiotics, and are therefore currently being proposed for different biomedical applications [11]. The integumentary system is the body’s protective system against microorganism invasion. When this barrier is disrupted, in the case of deep wounds or burns, therapeutic and pharmacological advances can improve the healing process and prevent microbial colonization and associated infections. Given the persistent emergence of drug-resistant microorganisms and the continued lack of new antibiotics, there is an urgent need to develop new modalities of bacteriostatic and bactericidal action [12]. AgNPs have drawn significant attention from the scientific community, prompting extensive research to elucidate their physical, chemical, and biological characteristics, mechanisms, and uses in both medical and non-medical fields. As a viable alternative and adjuvant to antibiotic therapy, nanoparticles are suitable to combat resistance to several drugs [13]. Due to their structure, concentration and unique ways of establishing contact with bacterial cell surfaces, a complete mechanism of action has not been explicitly elucidated. To date, numerous studies have described various mechanisms of action. One of them suggests that due to their size and shape, AgNPs provide a high surface area to volume ratio, which could allow nanoparticles to attach to the surface of a microbe, increasing its permeability and leading to membrane dissolution [14,15]. Increased permeability could also allow AgNPs to easily penetrate a microbe and damage intracellular organelles, including mitochondria and ribosomes, and disrupt biomolecules such as proteins and DNA [13,15,16,17]. This would cause cellular toxicity and oxidative stress by generating reactive oxygen species (ROS) and free radicals [18,19]. It is important to note that numerous studies have described the strong antibiofilm formation activity of AgNPs, including by disorganizing porin proteins, preventing glycocalyx formation and enhancing the effect of commonly prescribed antibiotics against Gram-positive and Gram-negative bacteria [20,21,22].

The antimicrobial action of AgNPs is further attributed to a complex interaction between size, shape and various reported optimal concentrations. Because their surface area to volume ratio is large, as AgNP size decreases, the bacteriostatic and bactericidal activity of the particles increase [17]. Investigations about the shape-dependent antimicrobial activity of AgNPs have included spherical, rod-shaped and triangular, with the last one receiving the most recognition for its antimicrobial properties [23].

Given the beneficial interactions of AgNPs with living structures and their nontoxic effects on healthy human cells, they represent an ideal candidate for various biomedical products and biomedicine [24]. Another study revealed that AgNPs significantly inhibited the viability of NIH3T3 mouse embryonic fibroblasts in a dose-dependent manner. Also, the cell proliferation test performed showed that AgNPs significantly inhibited cell proliferation, also in a dose-dependent cytotoxic manner [25]. A recent study assessed the viability of normal Human Skin Fibroblast (HSF) cell line using MTT assay after 24 and 48 h of exposure at different concentrations of the aqueous colloidal AgNPs. The results showed a significant concentration- and time-dependent manner in the reduction of cell viability. The cytotoxic effect of AgNPs was higher for cells after 48 h of exposure compared with 24 h. The results highlighted that small silver particles of nanometric size could have cytotoxic effects on normal cells at high concentrations and at a longer exposure time [26]. The general objective of this study was to develop new biocomposite films based on chitosan (CS) and poly (vinyl alcohol) (PVA) containing, on the one hand, AgNp coated with chitosan, and on the other hand, ibuprofen, with its dual antimicrobial and anti-inflammatory properties. The CS used for the coating of AgNps will have not only the purpose of stabilizing the nanoparticles, but also of assuring the compatibility between the metal and the film in the final composite material [27,28]. CS, a linear polysaccharide of randomly distributed N-acetyl glucosamine and glucosamine units, exhibits minimal foreign body reaction, controllable mechanical/biodegradation properties, biocompatibility, hydrophilicity, non-toxicity, non-antigenicity, and anti-microbial activity as well as bio-adherence and cell affinity [29]. PVA is a non-toxic polymer with good flexible and foldable film-making properties, all these properties of PVA making it a suitable agent for the preparation of flexible membranes for soft tissue engineering applications [30]. The original aspect of this study consists in combining the antimicrobial CS-coated AgNPs with CS/PVA films that proved to control the release of some small particles. The dual antimicrobial and anti-inflammatory properties of the new composite material will assure its applicability in local treatment of different diseases.

## 2. Results and Discussion

### 2.1. Characterization of the Obtained AgNps

The FTIR spectrum of AgNps coated with CS is shown in Figure 1. The presence of chitosan is demonstrated by the band around 3300 cm^−1^ which is specific to the amine and hydroxyl groups of chitosan. Some bands in the AgNPs-CS spectra such as 1647, 1572 and 1315 cm^−1^ respectively may be due to the interaction between silver and nitrogen atoms in the primary amine and amide groups [31].

The morphology and size of the AgNPs were determined by TEM and DLS. Figure 2a shows that the metallic nanoparticles have spherical shape with diameters between 4 and 22 nm. Figure 2b shows the size distribution curve in number for AgNps, and as can be seen, it has a monomodal distribution. The mean diameter of AgNPs was 7.37 nm (SD = 0.69; PDI = 0.62).

The nanoparticles have a content of Ag of 18 wt%, as determined from AAS measurements. CS from the surface of the nanoparticles (82 wt%) stabilize the AgNPs and assure a good homogenization in the film and the participation in the ionic and covalent crosslinking.

### 2.2. Structural Characteristics of Films

FT-IR spectra for films with and without AgNps, loaded or not loaded with IBF, are shown in Figure 3 and Figure 4.

Following the FTIR analysis, it was found that the incorporation of AgNps into the films led to a decrease in the intensity of several peaks in the spectrum of C2P2.10 and C2P4.10 samples, or even to their displacement. The peak at about 1590 cm^−1^ in the spectrum of samples without AgNps (C2P2.10 and C2P4.10) indicates the presence of amino groups (NH_2_) specific to the polysaccharide structure. In the spectra of the composite film containing AgNps, there is a decrease in the intensity of this peak, but also a displacement of it to about 1584 cm^−1^ which reflects the interactions between the atoms of Ag, O and N. Also, this peak can be attributed to the imine group (C=N) which proves the covalent crosslinking between GA and CS. The absorption bands located at approximately 1733 cm^−1^ and 1727 cm^−1^ which are found in the spectra of all analyzed samples (except in the IBF spectrum), can be associated with the stretching vibration of the –C=O group of PVA. The FTIR spectrum of IBF shows characteristic peaks at 1475 cm^−1^ and between 780 cm^−1^ and 634 cm^−1^, which can be associated with the vibration of aromatic groups. The appearance in the spectrum of C2P4.10-IBF and C2P4.10.Ag1-IBF films of peaks at 783 cm^−1^, 772 cm^−1^, 763 cm^−1^ and 646 cm^−1^ reveals that the drug was successfully incorporated into films [32].

### 2.3. Morphological Characteristics

The morphology of films with and without AgNps and of drug-free films was investigated by scanning electron microscopy (SEM), which is shown in Figure 5. Cross-sectional microscopy of C2P2.10 films reveals the presence of fibrillar formations attributed to polysaccharide. C2P4.10 type films have a porous structure that may be due to an increase in the amount of PVA in the system. In the case of films without AgNps (C2P4.10 and C2P4.10-IBF), the pores have an irregular shape and larger dimensions than in the case of films with AgNps (C2P4.10.Ag1 and C2P4.10.Ag1-IBF), which have a structure more homogeneous with smaller and more uniform pores. This change in film morphology is mainly due to the addition of AgNps which helps to fill the mesh of the polymer network. EDAX analysis revealed the presence of AgNps in the analyzed biocomposite films (Figure 6).

Elemental EDAX analysis revealed the presence of AgNp in C2P2.10.Ag1-IBF and C2P4.10.Ag1-IBF films (Figure 6). Following the calculations performed based on the EDAX analysis, it was found that the percentages of AgNp remaining in the films were approximately 60% in the C2P2.10.Ag1-IBF hydrogel, and approximately 50% in the C2P4.10.Ag1-IBF hydrogel. After the purification step, a certain amount of AgNp was removed from the network by the biocomposite films, thus explaining the decrease in the percentage of AgNp in the film compared to the initial one, this being also highlighted by UV-Vis spectroscopy (Appendix A).

### 2.4. Mechanical Properties of Films

The tensile properties of the obtained films in the dry state were investigated, and the stress-strain curves obtained are shown in Figure 7. 

The tensile strength varied from 185 to 430 KPa, both the presence of Ag nanoparticles and the drug having an influence on the mechanical properties of the films. Increasing the PVA content in the C2P4.10.Ag1 and C2P4.10.Ag1-IBF samples caused an increase in the strain at break. The obtained results are in agreement with the literature data showing that PVA can be used as a polymer to improve the mechanical properties of hydrogels [33]. C2P2.10 films had a lower breaking strain (C2P2.10.Ag1—10% strain; C2P2.10.Ag1-IBF—13% strain) than did C2P4.10 films (C2P4.10.Ag1—deformation 13%; C2P4.10.Ag1-IBF—deformation 17%).

### 2.5. Swelling Capacity of Films in Aqueous Solution

The results of the swelling degree are presented in Figure 8. It is known that most bacteria, including staphylococci, grow at a pH that varies between 5 and 8, and therefore, a pH of 6.8 was chosen for this study [34]. 

The swelling behavior of the analyzed films was influenced by the presence of AgNp in the films, but also by the ratio between the two polymers (CS and PVA). As expected, it was found that the films containing AgNp show a lower degree of swelling than the films without AgNp, a behavior that can be attributed to the decrease of the space in the meshes of the polymer network and, implicitly, to the increase in the density of the biomaterial. The C2P2.10 sample has a swelling degree 8% higher than that of the C2P2.10.Ag1 sample and 15% higher than the C2P2.10.Ag1-IBF sample. Also, the C2P4.10 shows a degree of swelling 8% higher than the C2P4.10.Ag1 sample and 6% higher than C2P4.10.Ag1-IBF. There is a greater difference in the swelling degree between C2P2 samples than between C2P4 samples. It was found that the decrease in the ratio between the two polymers led to a decrease in the swelling degree. The swelling degree in the environment with pH = 6.8 for the films type C2P2.10 varied between 35% and 50%, and for the films type C2P4.10 it was between 23% and 15%.

### 2.6. Films Loaded with IBF

The amounts of IBF loaded into obtained films are presented in Table 1. The obtained results evidenced that the amount of IBF loaded into the films containing AgNp was lower than those without AgNp. It was also observed that in films type C2P2.10 was found a slightly higher amount of ibuprofen compared to the films type C2P4.10, which is in accordance with the results of the swelling degree.

### 2.7. In Vitro Release of IBF

In vitro release profiles of simple IBF and also of IBF from the films, in phosphate buffer solution (PBS) (pH = 6.8), are presented in Figure 9. The release of the IBF from the analyzed films is controlled by the process of diffusion through the polymer membrane. The results showed that the re-lease efficiency of the simple drug was approximately 96% after 6 h from the start of the experiment, while the release efficiency of IBF from the films varied between 79% and 92% after 72 h from the start of the experiment. It was found that the C2P2.10 samples had a slightly higher release degree of IBF than the C2P4.10 films. This behavior is in concordance with the results of swelling degree.

### 2.8. The In Vitro Cytotoxic Effects

The cytotoxicity assessment provides the information needed to determine if the obtained material is biocompatible and can be used in biomedical applications. Human dermal fibroblastic cells (HDFa) were used as model cells for the evaluation of the in vitro cytotoxicity of the films (with and without the drug). The effect of the films on the viability of fibroblasts after 6 h and 24 h of incubation was evaluated using the MTT colorimetric test adapted according to the protocol described by Mosmann [35]. Cell viability was calculated based on mitochondrial function by reducing 3- (4,5-dimethylthiazol-2-yl) -2,5-diphenyltetrazolium bromide (MTT) to a colored insoluble formazane salt. The obtained results are presented in Figure 10.

The cell viability in contact with films without AgNPs and the drug was greater than 90% after 6 h and greater than 86% after 24 h, as can be seen from Figure 10. The addition of the drug or AgNPs has led to a decrease in cell viability. In the case of films with AgNPs, but without the drug, cell viability after 6 h of incubation was around 89%, and around 85% after 24 h of incubation. In the case of drug loaded films, without AgNPs, cell viability after 6 h of incubation was around 85% and around 80% after 24 h of incubation. Another small decrease in viability was recorded when both AgNPs and the drug were incorporated, with viability reaching 82% after 6 h of incubation and 79% after 24 h of incubation. However, cell viability values have not fallen far below the 80% limit, proving that the films tested are non-toxic and can be used for biomedical applications [36].

### 2.9. Evaluation of Antimicrobial Activity

Agar disk diffusion is a standardized method used in many microbiology laboratories for routine antimicrobial susceptibility testing [37]. Through this method, using specific culture media and different incubation conditions, the results can be interpreted by referring to the diameter of the inhibition zones. In Figure 11 are shown the images of the *S. aureus* (gram-positive bacteria) and *E. coli* and *K. pneumoniae* (gram-negative bacteria) agar plates over which the film discs were added. It can be seen that the antimicrobial agent (AgNPs) diffused into the agar and inhibited the germination and growth of the test microorganism. The diameters of the inhibition zones (d_iz_) were measured and with the obtained results the graph presented in Figure 12 was prepared. The d_iz_ was transcribed as “−” when no antimicrobial effect was noticed, “+” when the d_iz_ was <15 mm, and “++” when the d_iz_ was between 15 and 25 mm [38].

Four samples discs were placed on each agar plate: the sample without the drug and without AgNPs (C2P2.10 or/and C2P4.10) and samples with drug (C2P2.10-IBF and C2P4.10-IBF) or drug and AgNPs in different concentrations (C2P2.10.Ag0.4-IBF, C2P2.10.Ag0.6-IBF, C2P2.10.Ag1-IBF and C2P4.10.Ag0.4-IBF, C2P4.10.Ag0.6-IBF, C2P4.10.Ag1-IBF). 

In the case of gram-negative bacteria, no antimicrobial activity was demonstrated. Both CS and AgNPs have antimicrobial activity against *S. aureus* proven by studies in the literature [23,39]. Consequently, the film samples containing AgNPs show significant antibacterial properties against *S. aureus*, while the samples without AgNPs have no detectable antimicrobial activity (Figure 11). The anti-inflammatory drug also has a beneficial influence, which improves the antimicrobial effect in combination with AgNPs. It was also observed that as the concentration of AgNPs increases, the zone of inhibition also increases. Therefore, the highest antimicrobial activity against *S. aureus* was observed for C2P2.10.Ag1-IBF and C2P4.10.Ag1-IBF samples (the films with 1% AgNPs).

Similar results regarding the antimicrobial activity against *S. aureus* of AgNPs incorporated in polymer matrices were also observed in other research studies [31,40].

### 2.10. In Vivo Tests

The evolution was favorable at 24 h, when the healing process had already started; in the group exposed to ibuprofen solution, blood clots remained on the skin surface (Figure 13).

At 72 h most of the lesions showed a good evolution, except for the group exposed to IBF. The skin irritated with alcohol, 95% over which patches were applied, had a very good evolution, compared to those without the irritant. The best healing results were noticed after 7 days of the experiment in rabbits of the C2P4.10.Ag1-IBF group, where the complete regeneration of the epidermis and hairs was be observed. No scabs were observed compared to the skin exposed to IBF. The other lesions showed almost complete healing, but small crusts could still be observed. 

At the 7th day of the experiment, MHC II expression marked keratinocytes in the epidermis, basal layer, parabasal layer, perifollicular stem cells and perivascular stem cells. This labelling was recorded in all groups. A higher frequency of positivity was recorded in the groups also exposed to the irritant C2P4.10.Ag1 + S.D.I., C2P4.10-IBF + S.D.I., IBF + S.D.I. The positivity is favorable to rapid recovery.

Collagen I showed positive labelling in all groups, but with higher positivity in decreasing order from groups IBF, IBF + S.D.I., C2P4.10-IBF + S.D.I. Both collagen fibers and a small number of fibroblasts and keratinocytes in the basal and parabasal layers were labelled. The presence of a higher percentage of collagen I fibers is characteristic of a scar with prolonged evolution.

TNF was strongly positive in C2P4.10.Ag1 + S.D.I., C2P4.10-IBF + S.D.I., and C2P4.10. TNF⍺ in this case is an antiapoptotic marker that stimulates the recovery of injured tissues. 

Cox2 showed strong to moderate positivity in groups IBF + S.D.I., C2P4.10-IBF, C2P4.10.Ag1 + S.D.I. (Figure 14). Mechanical or chemical injury could induce COX-2, which was also observed in this experiment.

Myofibrillar populations during the repair process were studied using αSMA antibody. After 7 days of exposure, myofibroblasts were present in the granulation tissue of all groups, preferentially located in the medial and deep dermal areas and associated with the margins of lesions. Higher numbers of myofibroblasts were captured in C2P4.10.Ag1-IBF, including the one with irritant and the IBF + S.D.I. group. The presence of intense expression marks the important proximity of the injured margins for reepithelialization.

The film used had a healing-inducing effect. None of the groups showed the granulation stage after 7 days of the experiment. The irritant agent had a stimulating effect, all markers used showing a more intense expression. Complete skin recovery was observed in the group exposed to C2P4.10.Ag1-IBF.

Skin wound healing in adult mammals is a complex multistep process involving overlapping stages of blood clot formation, inflammation, re-epithelization, granulation tissue formation, neovascularization and remodeling. Re-epithelisation consists in the reappearance of a new epithelium on the surface of a wound. The cellular and molecular processes involved in the initiation, maintenance and completion of epithelisation are essential for successful wound closure. A variety of modulators are involved, including growth factors, cytokines, matrix metalloproteinases, cellular receptors and extracellular matrix [41,42].

Multiple cellular and molecular processes are involved in the initiation, maintenance and completion of epithelialization. Formation of the provisional wound bed matrix from an insoluble protein exudate occurs. Migration of epidermal keratinocytes from the cut edges and their proliferation occurs, which feeds the advancing and migrating epithelial tongue, followed by stratification and differentiation of the new epithelium. Reformation of the intact basement membrane area and repopulation with specialized cells that direct sensory functions, pigmentation and immune parameters occur. Regeneration of a functional epidermis depends on the reconstitution of the dermal-epidermal junction (DEJ), which anchors the epidermis to the dermis [43,44], and the terminal differentiation of keratinocytes into a protective cornified layer [45]. Reepithelization begins approximately 16–24 h after injury, during the proliferation phase, and continues during the second and third phases of the wound healing process [46,47]. In acute skin injury, neutrophils, monocytes and macrophages are recruited to the injury site when the barrier is disrupted [48]. Subsequently, keratinocytes become activated, exhibiting a change in their phenotype that is orchestrated by growth factors, chemokines and cytokines produced by keratinocytes and other skin cells. The activated phenotype is marked by changes in the cytoskeleton and cell surface receptors. Keratinocytes move into the healing wound by polymerization of cytoskeletal actin fibers into outgrowth and formation of new complex adhesion fibers. Integrins and syndecans, cell surface receptors that lack enzymatic activity, transmit intracellular signals by interacting with structural and signalling molecules. These events are accompanied by: expression of αvβ5, αvβ6 and α5β1 integrins; various proteases such as plasminogen and matrix metalloproteinases (MMPs); growth factor, growth factor receptors; cell surface proteoglycans; and ECM components such as laminin [42,49,50]. The responding intracellular signalling pathways activate transcription factors that regulate keratin gene expression. Migrating keratinocytes show upregulation of K6, K16 and K17 keratins, which are thought to increase the viscoelasticity of migrating cells [51,52,53].

The existence of a keratinocyte activation cycle in which cells are first activated by the release of interleukin (IL)- 1 has been proposed by Coulombe [54] and Freedberg et al. [50]. Subsequently, the activated state is maintained by autocrine production of proinflammatory and proliferative signals. K6 and K16 are markers of the active state. 

Induced by IL-1, tumor necrosis factor (TNF) can maintain keratinocytes in an activated state [55]. Signals from lymphocytes, in the form of interferon, induce K17 expression and keratinocyte contractility [56]. This allows keratinocytes to contract the fibronectin-rich provisional matrix (FN). Signals from fibroblasts, in the form of transforming growth factor (TGF), induce expression of K5 and K14, which makes keratinocytes return to basal phenotype and complete the matrix for the activation cycle. Activated keratinocytes are hyperproliferative and migratory, change their cytoskeleton, have increased levels of cell surface receptors and produce dermo-epidermal DEJ junction components. MMP-9, which is produced by migratory keratinocytes, degrades DEJ components, allowing keratinocytes to migrate over the wound [57]. Activated keratinocytes also produce paracrine signals to alert fibroblasts, endothelial cells, melanocytes and lymphocytes, and produce autocrine signals that target keratinocyte neighbors. These responses are essential for orchestrating the actions of surrounding keratin types in repairing injured tissue. In turn, injured cell types produce their own autocrine and paracrine signals, which modify the actions of activated keratinocytes [52]. In addition, fibroblasts migrate beneath the wound site to close the wound [51]. Once the wound area is covered, contact inhibition causes them to stop migrating and triggers keratinocyte differentiation into keratinizing squamous epidermal stratified cells [58]. Keratinocytes, the predominant cellular component of the epidermis, derive from epithelial stem cells (EpSCs), which are mainly located in the hair follicle (HF) protuberance and in the basal layer of the interfollicular epidermis (IFE) [59]. After epidermal injury, EpSCs in both the HF and IFE niches give rise to keratinocytes that migrate and re-epithelize the wound [60]. A slight increase in collagen deposition was observed in our experiment in the scarified areas covered by films, which is very important because the wound repair process depends on collagen biosynthesis, deposition and maturation. In addition, collagen deposits in wounds provide the tensile strength of the formed scars [61].

Cyclooxygenase (COX) is an enzyme essential for prostaglandin biosynthesis. PGE2 production is essential for skin wound healing. PG is formed by the combined action of phospholipase, which releases arachidonic acid (AA) from cell membrane phospholipids, and cyclooxygenase (COX), which converts AA to PG. Mechanical or chemical injury could induce COX-2; such insults have included removal of superficial epidermis or topical application of irritants such as phorbol 12-myristate 13-acetate (PMA) or 12-O-tetradecanoylphorbol-13-acetate (TPA). COX-2 has also been induced in fibroblasts and macrophages by IL-1 or lipopolysaccharide (LPS) [62]. Accordingly, COX-1 is thought to be involved in normal skin homeostasis, while COX-2 is important in various responses to skin insults such as injury. After injury, these proteins and COX-2 mRNA were predominantly expressed in the upper and basal layers of epidermal wound margins, which are composed of migratory and proliferative cells [63]. Previous reports have also demonstrated COX-2 immunolabeling in the basal layer of mouse epidermis after treatment with an irritant, in contrast to the absence of COX-2 immunolabeling in normal mouse skin [64]. These proteins and COX-2 mRNA have also been observed in endothelial cells of small vessels and fibroblast-like cells within granulation tissue. These findings suggest that an increased level of COX-2 expression after injury, especially in the early acute phase, may enhance cell migration and proliferation underlying re-epithelization and angiogenesis. These cells proliferate in response to injury and may also give rise to migrating epidermal cells. The characteristic pattern of COX-2 protein expression may therefore be related to the distribution of follicular and epidermal stem cells via MHC1 molecules, keratinocytes present antigens to memory CD8+ T cells, inducing cytotoxic defense and inflammatory cytokine production [65], particularly following IFN-γ stimulation. 

Antigen presentation via MHC2 molecules has been documented in a mouse skin model in which MHC2 interfollicular keratinocyte MHC2 causes TH1 cell clusters formation [66]. Keratinocytes express adhesion molecules such as ICAM-1, which, together with B7 costimulatory molecules, promote lymphocyte recruitment [67]. Keratinocytes obtained from skin samples from healthy donors expressed immune markers on their surface. Over 95% expressed MHCI; in contrast, less than 5% expressed HLA-DR or CD86, and a few expressed the costimulatory molecule CD40. The ability of keratinocytes to achieve antigen presentation to CD4+ T cells via HLA class II in vitro is therefore limited. On the other hand, keratinocytes showed immunosuppressive properties in vitro, reducing CD4+ T cell proliferation under allogeneic conditions. Although differences in the frequency of dividing CD4+ T cells varied between individual PBMC donors, keratinocyte-mediated inhibition was observed in all cases. Addition of IFN-γ and TNF-α to mimic an inflammatory context altered keratinocyte HLA marker and keratin markers. Keratinocytes notably overexpress HLA-DR, as also demonstrated in mesenchymal stem cells (MSCs) [68]. However, even under these inflammatory conditions, keratinocyte-mediated inhibition of CD4+ T cell proliferation was still observed.

Fibroblasts, fibrocytes and myofibroblasts that play critical roles in both early and late phase contribute to wound contraction, collagen deposition, and ultimately, skin fibrosis [69]. Collagens are components of extracellular matrix (ECM) proteins that are found in almost all eukaryotic organisms excepting plants and protozoa [70]. There are approximately 27 different types of collagen that have been identified, with type I collagen being the most predominant. Type I collagen is found in the connective tissues of vertebrates, such as tendons, ligaments, bones, skin and cornea [71].

Tumor necrosis factor-alpha-inducing protein family 8 (TNFAIP8) have important roles in immune homeostasis, inflammatory responses, tumor genesis and development, and cell signal transduction. TNFAIP8, a 21kD cytoplasmic protein, has been reported to be expressed in both the cytoplasm and cell nucleus in human prostate cancer cells [72]. Most normal human tissues express TNFAIP8, which is highly expressed in immune-related tissues such as the lymphatic system, spleen, thymus, thyroid, bone marrow and placenta, while solid organs including ovary, kidney, heart, brain, testes and skeletal muscle show relatively lower TNFAIP8 expression. Carcinoma tissues can usually overexpress TNFAIP8. In addition, TNFAIP8 has been observed to be enriched in the kidney, brain and spleen of rats [73].

It has been indicated that TNFAIP8 appears to be involved in TNF-α-induced cell apoptosis. TNF-α can bind to tumor necrosis factor receptor 1 (TNFR1), which then enhances TNFAIP8 expression through activation of nuclear factor (NF)-κB. Increased TNFAIP8 expression and activity significantly inhibits caspase cascades, leading to decreased cell apoptosis [74]. In our study, the group treated with C2P4.10.Ag1-IBF showed the most intense expression among the experimental groups but also the best healing activity.

Autophagy is another pathway by which TNFAIP8 influences cell survival/death. TNF α treatment increased TNFAIP8 expression, which was associated with increased autophagy and decreased apoptosis [74].

A moist wound environment is thought to promote re-epithelization [75]. Wounds exposed to air lose water, the upper dermis dries out, and healing occurs under a dry crust. Covering a wound with an occlusive dressing prevents the appearance and formation of crusting and radically alters the healing pattern of epidermal wounds. When the wound is kept moist and dehydration is prevented by occlusion or semi-permeable membranes such as thin, transparent polyurethane adhesives, these wounds will re-epithelize faster than wounds that are allowed to dehydrate. This effect appears to be somewhat limited, as there appears to be a window of opportunity for the favorable use of moist wound occlusion [76]. Modern dressings have been developed to create and maintain a warm and moist environment, providing optimal conditions for enhanced healing [77], whereas traditional dressings can absorb a large amount of exudate, drying out the wound bed. Modern dressings are created from natural or synthetic polymeric materials or a combination of both, are available as thin films, foams or gels, and can be classified as hydrocolloid dressings, alginate dressings or non-alginated dressings.

Histological examination of wounds revealed that the composite drug formulations improved epidermal differentiation and multi-layered structure, which would be a consequence of better vascularisation of the regenerated tissue. Another example was the use of silver-impregnated dressing on deep burn wounds in humans [78]. Silver is an antiseptic agent. The study showed that exposure of a mesh skin graft to silver significantly increased the rate of mesh closure compared to a standard antibiotic solution. However, the mechanism is unknown as the pro-healing effect was not related to the antimicrobial properties of silver [79]. In superficial wounds, injuries are generally confined to the epidermis, and moisture retaining dressings (occlusive or semi-occlusive) are used to help promote re-epithelization. The main dressings for superficial wounds are hydrocolloids, hydrocellular dressings, metalloprotease inhibitors and hyaluronic acid (HA) based dressings. Superficial wounds, including thin burns, catheter insertion sites, partial-thickness wounds, and epidermal skin graft harvest sites, often require practical dressings known for their biocompatibility, biodegradability, and non-toxic nature and generally derived from natural tissues or artificial sources, such as collagen [80], HA [81], chitosan [82], alginate, and elastin. Polymers of these materials are used alone or in combination, depending on the type of wound. In general, dermal substitutes are preferred for full-thickness wounds and have been shown to minimize hypertrophic scarring and contractures and increase scar elasticity in acute burn wounds. They are also known to promote granulation tissue formation and accelerate endothelial cell migration and may consequently provide favorable conditions for re-epithelization. Biological wound dressings are sometimes incorporated with growth factors and antimicrobials to enhance the wound healing process.

## 3. Materials and Methods

### 3.1. Materials

Chitosan medium molecular weight (CS), ~75% degree of deacetylation (M.W.: 100,000–300,000 g/mol) and Poly (vinyl alcohol) (PVA), 87–89% degree of hydrolysis, M.W.: 13,000–23,000 g/mol were purchased from Acros Organics BVBA (Geel, Belgium). Glutaraldehyde 50% in aqueous solution (GA), Magnesium sulfate (anhydrous ≥98.0%) (MgSO_4_) and Glycerine ≥99.7% were acquired from VWR International. Human dermal fibroblasts cell line (HDFa), Dulbeco’s modified Eagle’s Medium (DMEM), 10% fetal bovine serum (FBS), antibiotics (streptomycin/penicillin), non-essential amino acids, phosphate-buffered saline (PBS) and trypsin-EDTA, used for in vitro cytotoxicity tests have been acquired from Thermo Fisher Scientific (Waltham, MA USA). 3-(4,5-Dimethyl-2-thia zolyl)-2,5-diphenyl-2H-tetrazolium bromide was obtained from Merck Millipore (Darmstadt, Germany). Freeze-dried stain (*Staphylococcus aureus*—ATCC 25923, *E. coli*—ATCC 11775, *Klebsiella pneumoniae*—ATCC BAA-1705) was purchased from ATCC (Manassas, VA, USA). Chapman agar was acquired from Oxoid (Hampshire, UK). For in vivo tests were used: xylazine, ketamine 10% formalin solution, ethyl alcohol, xylene, paraffin, CMH II (Dako M0746), anti\Cox2 (ab16701 SP-21), anti\TNFIP8 (ABIN 2707009), anti\Col I (abcam, ab 34710), anti\-SMA (Anti-alpha smooth muscle Actin antibody MA5-11547 (14A-asm-1), citrate acid buffer pH6, hydrogen peroxide, PBS, the goat anti-mouse IgG and the goat anti-rabbit IgG secondary antibody, 3,3’-diaminobenzidine (DAB) and hematoxylin.

### 3.2. The AgNps Preparation Method

Silver nanoparticles were obtained according to the method described by Popescu et al. [31]. Briefly, a AgNO_3_ solution (80 mM) was added to a CS solution in acetic acid (0.5% CS), so that the gravimetric ratio between AgNO_3_ and CS was 1.1:1. The mixture solution was stirred and kept at 90 °C for 18 h. The obtained AgNps covered with CS were separated by precipitation with NaOH until the solution pH was around 9. The precipitate was separated, purified by dialysis to remove Ag+ and other ions, and then recovered by freeze-drying.

### 3.3. The Films Preparation Method

CS and PVA based films were obtained by the double cross-linking method described by Rata et al. [38], according to the experimental program shown in Table 2. The mixture of the two polymers (PVA and CS) was covalently crosslinked using glutaraldehyde (GA). The reaction with GA took place both at the amine groups located along the CS chain, with the formation of imine bonds, and at the hydroxyl groups of PVA with the formation of new acetal bonds. Ionic crosslinking was performed using magnesium sulfate. This reaction involves the -NH_2_ type functional groups, in the form of the ammonium ion (belonging to chitosan) with the SO_4_^2−^ groups of the ionic crosslinker. Specific amounts of CS, AgNp coated with CS and PVA were placed in 50 mL 2% (*w*/*v*) lactic acid solution and left to stir overnight in the dark. Silver nanoparticles were obtained according to the method described by Popescu et al. [31]. The next day, a certain amount of 5% GA necessary to cross-link the free amino groups of the CS and the hydroxyl groups of the PVA was added to the polymer solution under strong magnetic stirring. After 30 min the MgSO_4_ solution was added keeping the magnetic stirring for 15 min. To obtain the films loaded with drug (C2P2.10-IBF, C2P2.10.Ag1-IBF, C2P4.10-IBF and C2P4.10.Ag1-IBF), the same steps were followed and after the addition of the two crosslinkers were introduced 5 mL of Ibuprofen solution with a concentration of 40 mg/mL. Finally, 500 mg of glycerin was added to the polymer solution. The mixture obtained was then carefully transferred to a square silicone shape with side of 6.5 cm. For drying, the samples were placed in the oven at a temperature of 37 °C. After drying the hydrogels were purified by placing them in Petri dishes over which a volume of 20 mL of ultrapure water was added. The films were kept in this medium for 2 min after which the medium was replaced. This procedure was repeated 3 times. A higher number of washes lead to the film breaking. After washing, the films were placed in perforated polypropylene bags and placed in filter paper bags until completely dry.

### 3.4. Characterization of the Obtained AgNps

The structure of AgNps was analyzed by FTIR spectroscopy using a Shimadzu IRSpirit spectrophotometer in ATR mode (400–4000 cm^−1^). Transmission electron microscopy (TEM) images were obtained using a Hitachi High-Tech HT7700 microscope (Tokyo, Japan) at 100 kV and the AgNPs-CS solution (1 mg/mL) was deposited onto 300 mesh carbon coated copper grid and dried under vacuum. The average diameter of AgNps was investigated by Dynamic Light Scattering (DLS) measurements in ddH2O using a Malvern Zetasizer Pro (Malvern Pananalytical, Worcestershire, UK). The silver content from AgNps-CS (dispersed in 5% HNO_3_ aqueous solution) was determined by atomic absorption spectroscopy (AAS) using a ContrAA 800 spectrometer from Analytik Jena (Jena, Germany) with air/acetylene flame at 328 nm.

### 3.5. Characterization of the Obtained Films

Obtained films with and without AgNps, loaded or not loaded with drug (IBF) were characterized from several points of view: structural, morphological, the degree of swelling in aqueous solutions, and the degree of loading and release of IBF. Also, the obtained films were evaluated in terms of antimicrobial activity, cytotoxicity and the healing effect of the films on the skin of rabbits.

#### 3.5.1. The Structural Characteristics

The structural analysis of the obtained films, and also of simple ibuprofen was performed by FTIR spectroscopy using a spectrophotometer type IRSpirit from Shimadzu, in the ATR mode (400 to 4000 cm^−1^).

#### 3.5.2. The Morphological Characteristics

Obtained films were analyzed in dry state by scanning electron microscopy (SEM) using an equipment type Quanta 200 Scanning Electron Microscope. The elemental composition of the biocomposite films and the presence of AgNp were evaluated using a Verios G4 UC Scanning Electron Microscope (Thermo Scientific) equipped with an EDAX module.

#### 3.5.3. Mechanical Properties of Films

Tensile strength of the films in dry state was evaluated using a TA.XT Plus Texture Analyzer, at room temperature according to the method described by Rata et al. [38]. For the test were used rectangular films with dimensions of 50 mm/10 mm (length/width) and were caught between clamps that were positioned at 20 mm distance. The films thickness was approximately 0.5 mm and the speed of the tensile measurements was of 0.5 mm/s. 

#### 3.5.4. Swelling Capacity of Films in Aqueous Solution

The maximum degree of swelling after 24 h of the films with and without silver nanoparticles and with and without the drug was evaluated in a PBS solution with pH = 6.8. The swelling degree of the obtained films was determined by the gravimetric method. Square pieces with a weight of approximately 200 mg of each type of hydrogel were weighed and immersed in 20 mL of PBS (pH = 6.8). The samples were left to swell at a temperature of 37 °C for 24 h under a continuous magnetic stirring of 50 rpm. After 24 h, the samples were removed from the aqueous environment and the excess water was removed by lightly tamponage of the films with filter paper. The degree of swelling of the samples was determined using the following equation:(1)$$Q\left(\%\right)=\frac{W_{1}-W_{0}}{W_{0}}\times 100$$
where, *W*_1_ is the weight of swollen film and *W*_0_ the initial amount of films in dry state.

#### 3.5.5. Films Loaded with IBF

The films were loaded with IBF during the synthesis process. To determine the amount of ibuprofen incorporated into the films, pieces of approximately 200 mg of each type of hydrogel were weighed and immersed in 20 mL of NaOH 0.05 N under continuous magnetic stirring (120 rpm) for 72 h, until complete solubilization of the drug. Every 24 h the NaOH solution was refreshed and the amount of IBF in the resulting supernatant was quantified by UV-Vis spectrometry (using a Nanodrop One, Thermo Scientific) at 223 nm. The experiment was performed in triplicate using square samples from different parts of the hydrogel. The 0.05 N NaOH solution was used to favor the breaking of NH_3_^+^COO^−^ bonds between IBF and CS that can form when the drug is added to the system. The amount of IBF loaded into the films was calculated using the following equation:*m*_l_ = *m*_i −_
*m*_s_(2)
where, *m*_l_: the amount of IBF loaded in films (mg), *m*_i_: the initial amount of IBF (mg), *m*_s_: the amount of IBF found in the supernatant (mg).

#### 3.5.6. In Vitro Release of IBF

To evaluate the release degree of IBF from films, a phosphate buffer solution (PBS) with a pH of 6.8 was used. This experiment was carried out using a dissolution apparatus (Dissolution Apparatus 708-DS) provided with a sampling station (Sampling Station, 850-DS). For this test, rectangular samples of each type of hydrogel with a weight of approximately 200 mg were used, which were inserted into dialysis tubes over which 1 mL of phosphate buffer solution with a pH of 6.8 was added. The samples thus prepared were introduced into a volume of 200 mL. The experiment was conducted for 72 h, at a temperature of 37 °C under a continuous stirring of 50 rpm. At predetermined time intervals, a volume of 500 μL of supernatant was automatically taken and replaced with the same volume of fresh PBS solution. The drug concentration in the supernatant was determined by UV-Vis spectroscopy (Nanodrop One, Thermo Scientific) at 223 nm. IBF release efficiency (*Ref* %) was calculated using Equation (3):(3)$$Ref\left(\%\right)=\frac{mr}{ml}\times 100$$

#### 3.5.7. The In Vitro Cytotoxic Effects

The in vitro cytotoxic effects of biocomposite films were evaluated using Human Dermal Fibroblasts, adult (HDFa). HDFa cells were cultured and prepared for tests, as described in detail in our previous work [38,83]. The film samples, previously sterilized, were cut with a biopsy perforator (to obtain discs with a diameter of 3 mm), and incubated with HDFa cells. The mass of the disc was ~ 4 mg. After the incubation time (6 h and 24 h), the cytotoxic effect was determined using the MTT assay. All necessary procedures were performed in a Kartusalan Lamil Plus 13 hood with laminar flow. Each film sample was tested in triplicate to obtain the average percentage and standard error.

#### 3.5.8. Evaluation of Antimicrobial Activity

The antimicrobial activity of the obtained films against the typical *S. aureus*, *E. coli* and *K. pneumoniae* reference strains was determined using the disc diffusion technique, as well as in other studies undertaken by our research team [38]. Commercially available antimicrobial discs with Chapman agar (gram-positive bacteria) or MacConkey agar (gram-negative bacteria) were used for these tests. A suspension of microorganisms (0.5 McFarland density) was inoculated on a Petri dish with the substrate. The sterilized film discs (4 mm diameter) were first hydrated in ultrapure water and then placed on the agar plate and incubated for 24 h at 37 °C. Antimicrobial activity was evaluated by measuring the diameters of the inhibition zone according to the Kirby-Bauer method [84].

#### 3.5.9. In Vivo Tests

The in vivo testing was carried out at the Experimental Animal Unit of the University of Life Sciences “Ion Ionescu de la Brad”, and in the Histology Laboratory.

The objective of this research was focused on the study of the healing effect of some films (C2P4.10, C2P4.10-IBF, C2P4.10.Ag1, C2P4.10.Ag1-IBF compared to IBF) on the skin of rabbits submitted to the experiment.

The biological material was represented by 25 rabbits, common breed, with an average age of 1 year, divided into 5 experimental groups of 5 animals each. The animals were housed and handled in strict accordance with the European Animal Welfare Act, under the supervision of veterinarians and were monitored for evidence of disease and changes in attitude, appetite or behavior suggestive of disease.

Ethical approval for the study was obtained from the Ethics Committee of the Faculty of Veterinary Medicine, University of Life Sciences “Ion Ionescu de la Brad” from Iași (no. 181/01.03.2021).

The rabbits included in the experiment were homologous in terms of breed, age, body weight and health status. They were housed in the same room with similar environmental conditions (temperature, humidity, lighting) and the same food. The animals were sedated by intramuscular injection of xylazine and ketamine, then the skin was scarified in two areas/animal, 1.5/1.5 cm each, on the dorsal thoracic side. The healing effect of the therapeutic formulas was monitored on one of the areas. The other area was irritated with 0.5ml of 95% ethyl alcohol and then patches with the therapeutic formulas were applied to monitor the healing effect. The patches with therapeutic formulas were changed every 24 and 72 h. Throughout the experiment, the rabbits were monitored and evaluated for health status. At the 7 day mark of the experiment the animals were sedated by intramuscular injection of xylazine and ketamine and then euthanized. Skin samples were prevealed from each treated location, then fixed with 10% formalin solution for 24 h, dehydrated with ethyl alcohol, clarified with xylene, paraffin embedded, sectioned at 4 µm, HE and IHC stained.

The immunohistochemical staining was made using anti\CMH II (Dako M0746), anti\Cox2 (ab16701 SP-21), anti\TNFIP8 (ABIN 2707009), anti\Col I (abcam, ab 34710), and anti\-SMA (Anti-alpha smooth muscle Actin antibody MA5-11547 (14A-asm-1).

After dewaxing the sections in Xylen, hydrating in ethanol and microwaving for 10 min at 95 °C in 10 mmol citrate acid buffer pH6, they were cooled for 20 min, and after that twice washed in PBS for 5 min. The slices were treated with 3% hydrogen peroxide, and rinsed with PBS, after that being incubated overnight at 4 °C in humid atmosphere with primary antibodies, under dilution of 1:100 TNFIP8, Col I, CMH II, Cox2 and 1:800 for ⍺-SMA. The next day, the slides were 3 times washed in PBS for 5 min being incubated with the secondary antibodies. The goat anti-rabbit IgG secondary antibody was used to reveal TNFIP8, CMH II, Cox2,Col I activity of sckin cells and for, ⍺-SMA, the goat anti-mouse IgG secondary antibody were chosen. The microscope slides were developed in 3,3’-diaminobenzidine (DAB) before being finally counter stained with hematoxylin.

## Figures and Tables

**Figure 1 ijms-23-10671-f001:**
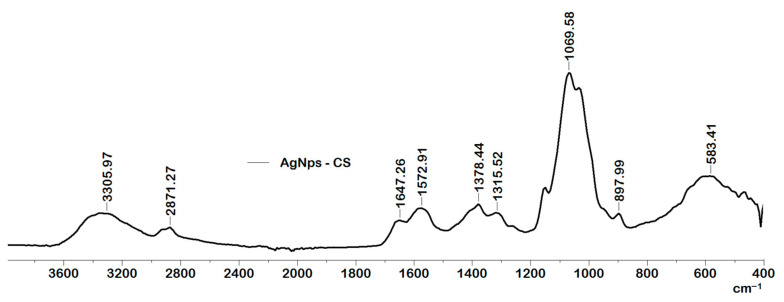
FT-IR spectra for AgNps covered with CS.

**Figure 2 ijms-23-10671-f002:**
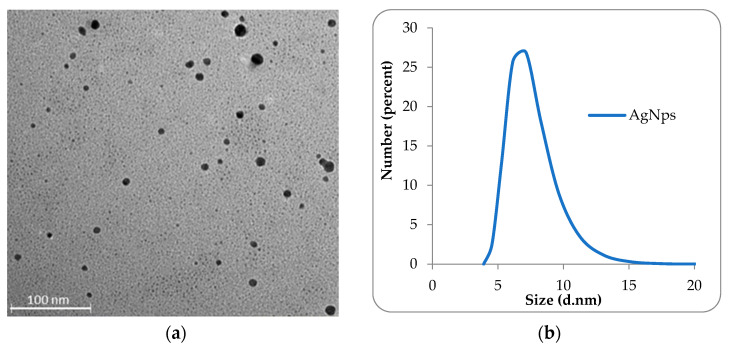
TEM image (**a**) and size distribution curve in number for AgNps (**b**).

**Figure 3 ijms-23-10671-f003:**
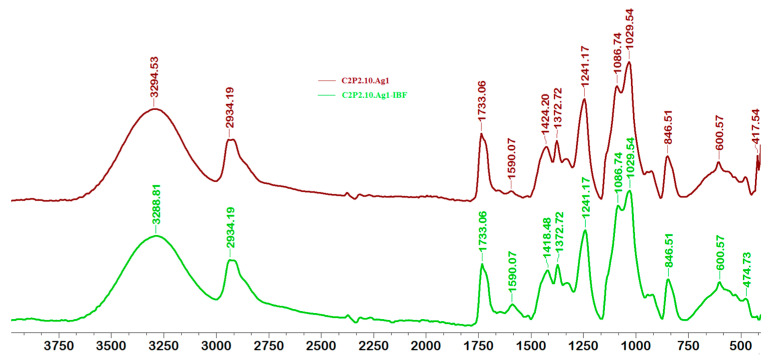
FT-IR spectra for films type C2P2.

**Figure 4 ijms-23-10671-f004:**
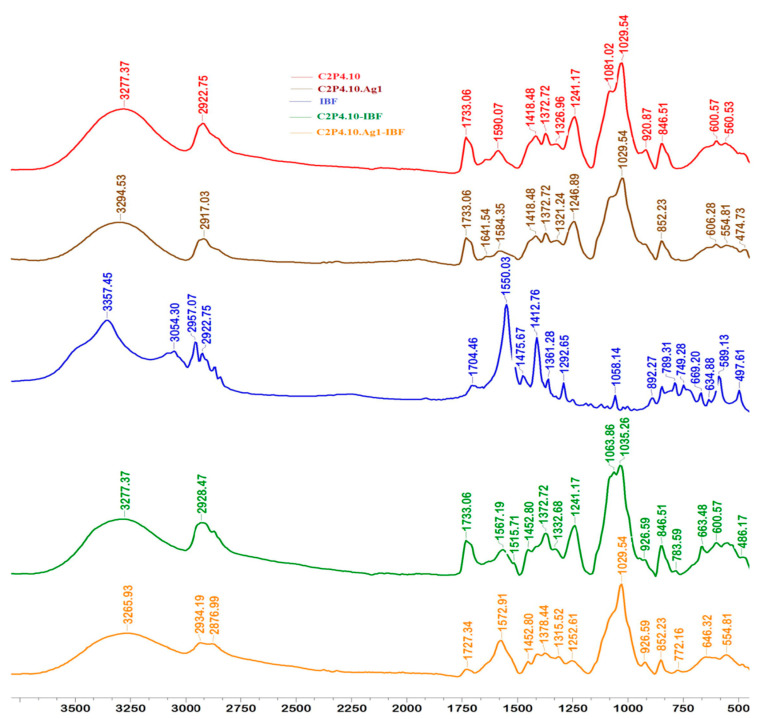
FT-IR spectra for films type C2P4.

**Figure 5 ijms-23-10671-f005:**
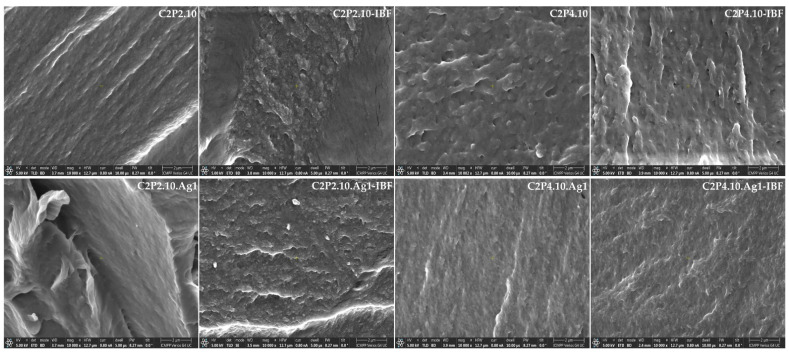
SEM micrographs in cross section of the obtained films.

**Figure 6 ijms-23-10671-f006:**
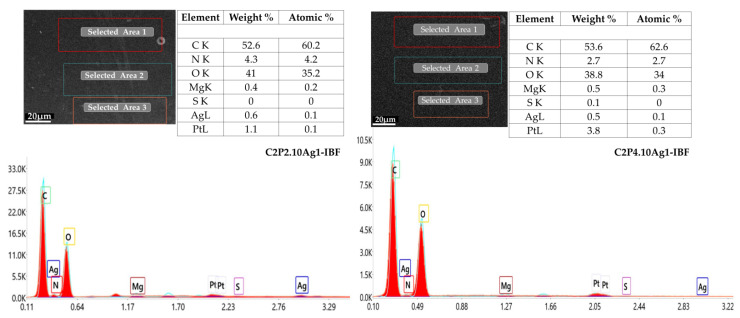
EDAX spectrum of composite films (scale bar-20 µm).

**Figure 7 ijms-23-10671-f007:**
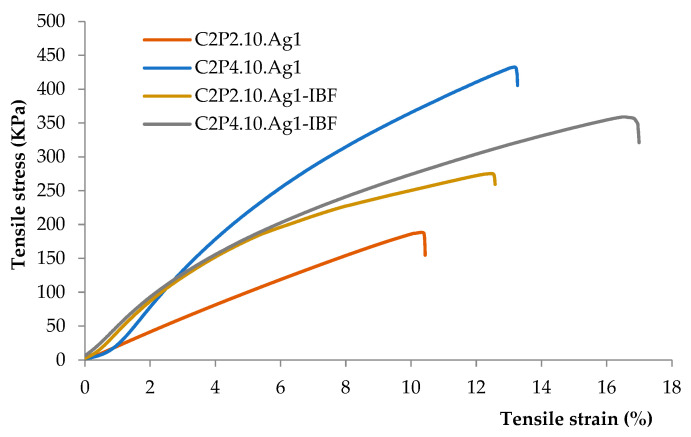
Stress-strain curves for series C2P2.10.Ag1 and C2P4.10.Ag1.

**Figure 8 ijms-23-10671-f008:**
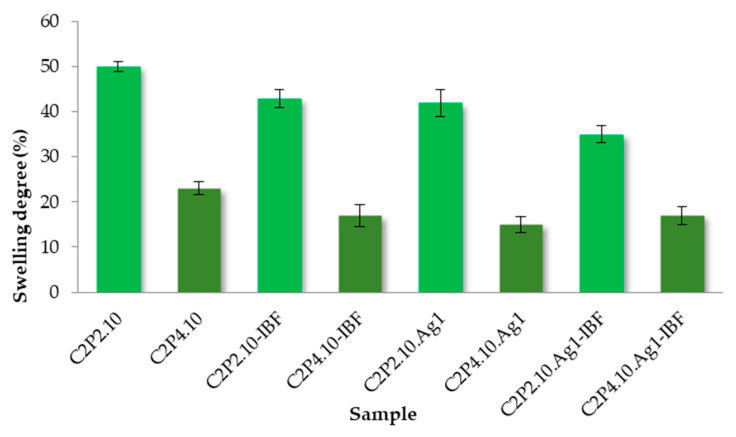
The swelling degree of the films after 24 h in PBS (pH = 6.8).

**Figure 9 ijms-23-10671-f009:**
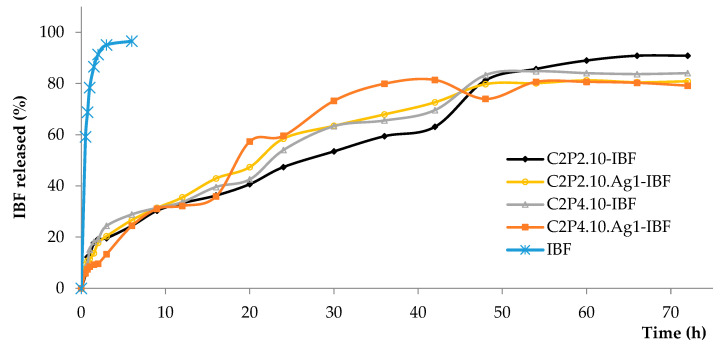
In vitro release kinetics of simple IBF and IBF from films in PBS (pH 6.8).

**Figure 10 ijms-23-10671-f010:**
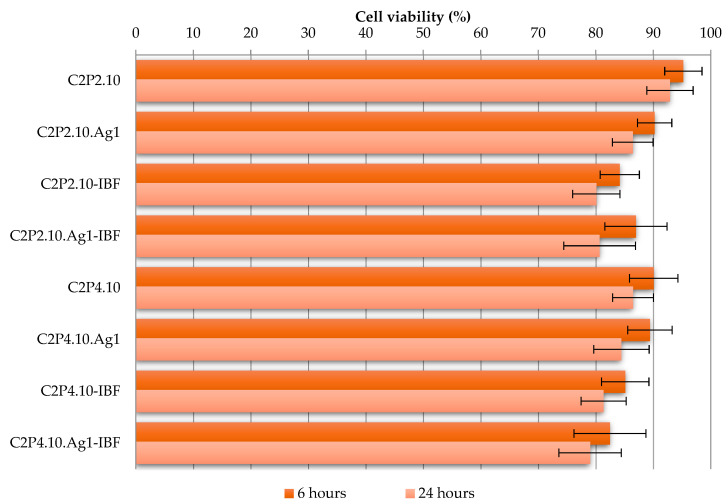
In vitro cells viability of obtained films at 6 h and 24 h after incubation.

**Figure 11 ijms-23-10671-f011:**
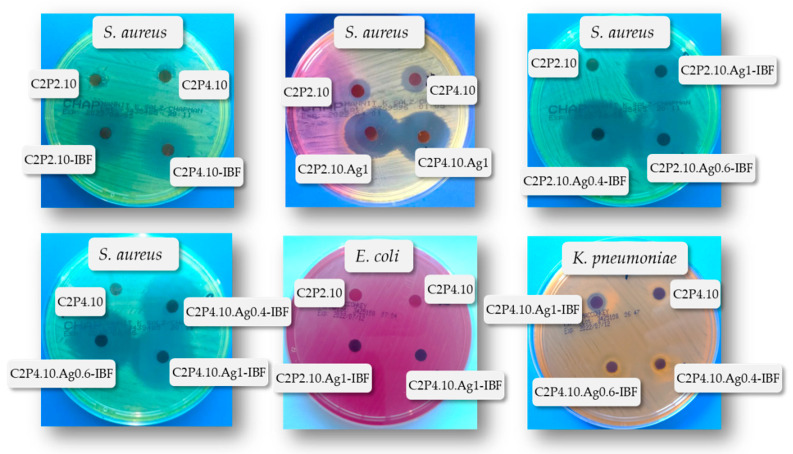
Antibacterial activity against *S. aureus*, *E. coli* and *K. pneumoniae* of film samples with and without the drug and with different percentages of AgNPs in the composition.

**Figure 12 ijms-23-10671-f012:**
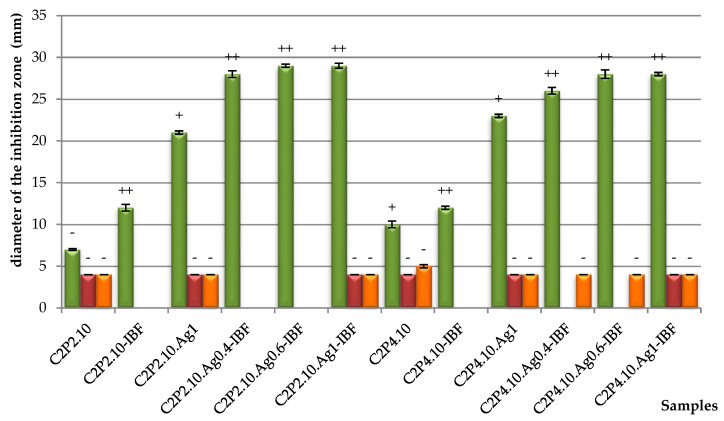
Diameter of inhibition zone of bacteria against *S. aureus (green bars)*, *E. coli (red bars)* and *K. pneumonia (orange bars)* obtained after 24 h of incubation at 37 °C in the presence of film samples with and without the drug and with different percentages of AgNPs in the composition.

**Figure 13 ijms-23-10671-f013:**
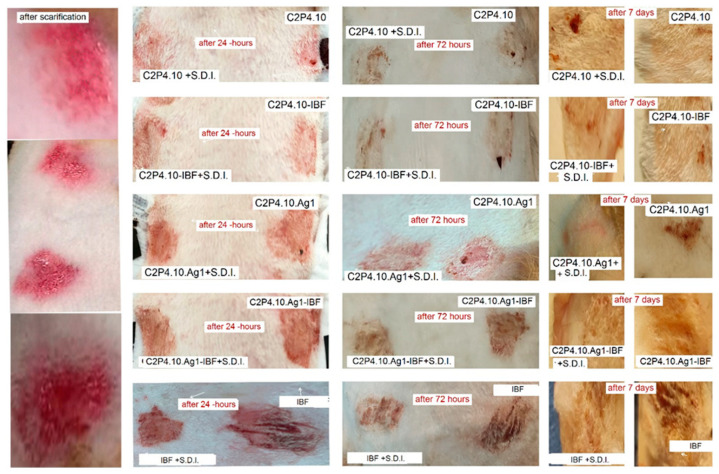
Macroscopic appearance of scarified skin exposed to patches of various drug formulations. A very good evolution can be observed at 7 days in groups treated with formulas C2P4.10.Ag1-IBF simple and C2P4.10.Ag1-IBF +S.D.I. and C2P4.10-IBF and less favorable in those with IBF. S.D.I. (Solution Dermal Irritations).

**Figure 14 ijms-23-10671-f014:**
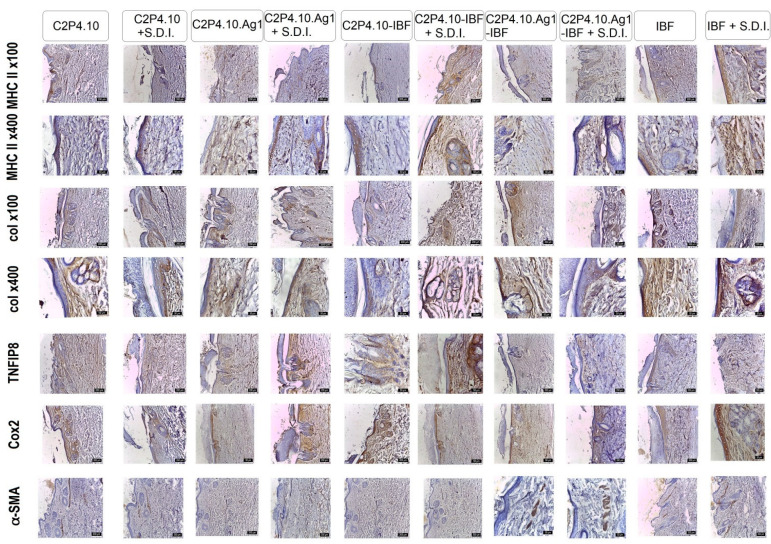
Col IHC of skin from rabbits from experimental and control groups, on 7th day of the experiment, exposed to different therapeutic combinations for healing with or without irritant. Col IHC with anti-CMH II, anti-Cox2, anti-TNFIP8, anti-Col I, anti-SMA.

**Table 1 ijms-23-10671-t001:** The amount of IBF loaded into 1 g films.

Sample Code	The Amount of IBF Loaded in Films (g IBF/g Film)
C2P2.10-IBF	0.063
C2P2.10.Ag 1-IBF	0.057
C2P4.10-IBF	0.061
C2P4.10.Ag1-IBF	0.059

**Table 2 ijms-23-10671-t002:** Experimental plan used for films preparation.

Sample Code	CS/PVA Ratio (mg/mg)	Moles of GA/Moles of Free NH_2_ and OH	Moles of MgSO_4_/Moles of Free NH_2_	AgNps in Relation to the Amount of Polymers (%)	Ibuprofen (mg)
C2P2.10	50/50	1/10	1/20	-	-
C2P2.10.Ag1	50/50	1/10	1/20	1	-
C2P2.10-IBF	50/50	1/10	1/20	-	200
C2P2.10.Ag1-IBF	50/50	1/10	1/20	1	200
C2P4.10	50/100	1/10	1/20	-	-
C2P4.10.Ag1	50/100	1/10	1/20	1	-
C2P4.10-IBF	50/100	1/10	1/20	-	200
C2P4.10.Ag1-IBF	50/100	1/10	1/20	1	200

## Data Availability

The data presented in this study are available on request from the corresponding author.

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
