# Peer review of "Silver Nanoparticles Biocomposite Films with Antimicrobial Activity: In Vitro and In Vivo Tests"

_ijms, 2022, doi:10.3390/ijms231810671_

Round 1

Reviewer 1 Report

This article present synthesis and characterizations of silver nanocomposites with chitosan. This study will help to facilitate to understand synthesis mechanism of nanoparticles with organic polymers. Before recommending this article for publication, there are some shortcomings for that should be resolve.

General comments

Overall, the study is well designed and presented in a good way, but mostly the literature is not cited. Grammatical and typos must be revised

Abstract

Methods are not well presented in the abstract

Which characterizations were performed to study the structure and morphology of the synthesized NPs.  

Also add quantitative results in this section.

Sentence first could be “maximum use/ genetic modifications in bacteria due to antibiotics.

The authors are directed to start the first sentence of the abstract with proper and concrete sentence.

What is the benefit of this study? As many studies have already presented the same techniques.

Introduction

The introduction part is well written but still some details are required. The authors should provide details of the NPs, its antibacterial and antimicrobial properties, physical and chemical properties of the AgNPs.

Add reference in line 67. The following articles could be helpful.

 https://doi.org/10.1016/j.bcab.2020.101729, https://doi.org/10.1007/s10534-022-00417-1,

Briefly discuss toxic behavior of the AgNPs.

Add information of the biological properties of the AgNPs

Materials and methods

Methodology is well written. Write cm-1 like this

Check and correct chemicals names and its formulas

Results

Present XRD and UV vis spectroscopy results.

Figure 6 add error bars and also perform statistical analysis to find significant difference

Discussion

Compare the obtained results with more current study.

Discussion must be more elaborative. It would be better to discuss every result and compare with current studies

Conclusion

Conclusion is well justified.  

Author Response

Response to Reviewer 1 Comments

Point 1: This article present synthesis and characterizations of silver nanocomposites with chitosan. This study will help to facilitate to understand synthesis mechanism of nanoparticles with organic polymers. Before recommending this article for publication, there are some shortcomings for that should be resolve.

Response 1: Thanks for your appreciation and suggestions.

Point 2: General comments

Overall, the study is well designed and presented in a good way, but mostly the literature is not cited. Grammatical and typos must be revised

Response 2: The manuscript was grammar and spell-checked. The list of bibliographic references was also improved

Point 3: Abstract

Methods are not well presented in the abstract.

Response 3: The abstract was modified taking into account the indications received. Considering that the limit is 200 words, not many details could be entered.

Point 4: Which characterizations were performed to study the structure and morphology of the synthesized NPs.

Response 4: The structure of the silver nanoparticles was evaluated by FTIR spectroscopy and the morphology by TEM microscopy

Point 5: Also add quantitative results in this section

Response 5: The Ag content in the nanoparticles was determined from AAS measurements

Point 6: Sentence first could be “maximum use/ genetic modifications in bacteria due to antibiotics. The authors are directed to start the first sentence of the abstract with proper and concrete sentence.

Response 6: The first sentence of the abstract has been improved.

Point 7: What is the benefit of this study? As many studies have already presented the same techniques.

Response 7: The benefit of this study compared to other studies in the literature was discussed in the Introduction. The number of words in the abstract is limited.

In the specialty literature have been reported the obtaining of hydrogels based on chitosan (CS) and poly (vinyl alcohol) (PVA) containing silver nanoparticles by the freezing-thawing cycle [D.M Suflet, I. Popescu 1, I.M. Pelin, D.L. Ichim, O.M. Daraba, M. Constantin, G. Fundueanu, Dual Cross-Linked Chitosan/PVA Hydrogels Containing Silver Nanoparticles with Antimicrobial Properties, Pharmaceutics, 2021, 13, 13(9):1461 Kumar, A., Behl, T., Chadha, S. Synthesis of physically crosslinked PVA/Chitosan loaded silver nanoparticles hydrogels with tunable mechanical properties and antibacterial effects. Int. J.Biol. Macromol., 2020, 149, 1262-1274] but at the best of our knowledge, there are no studies on the obtaining of this type of biocomposite film using the solvent casting method. The greatest benefit of these types of biocomposite films is that they can possess both antimicrobial (due to AgNPs) and anti-inflammatory (due to ibuprofen) properties. Also, the using of the double crosslinking method has been choosing to increase the stability of the biocomposite film and to reduce the quantity of the potential toxic covalent crosslinking agents.

Point 8: Introduction

The introduction part is well written but still some details are required. The authors should provide details of the NPs, its antibacterial and antimicrobial properties, physical and chemical properties of the AgNPs.

Response 8: The introduction was improved according to the reviewer's suggestions.

Point 9: Add reference in line 67. The following articles could be helpful.

 https://doi.org/10.1016/j.bcab.2020.101729, https://doi.org/10.1007/s10534-022-00417-1,

Response 9: The reference has been added

Point 10: Briefly discuss toxic behavior of the AgNPs.

Response 10: Toxic behavior of the AgNPs was briefly discussed.

Point 11: Add information of the biological properties of the AgNPs

Response 11: The information of the biological properties of the AgNPs was added according to the reviewer's suggestions.

Point 12: Materials and methods

Methodology is well written. Write cm-1 like

Response 12: The change was made in manuscript

Point 13: Check and correct chemicals names and its formulas

Response 13: The suggested corretions were made in the manuscript

Point 14: Results

Present XRD and UV vis spectroscopy results.

Response 14: In our research laboratory we do not have equipment for XRD analysis and our collaborators from other institutions were on vacation in August, which made it impossible for us to carry out this analysis.

The results from UV-Vis spectroscopy have been added in the Supplementary Materials

 Point 15: Figure 6 add error bars and also perform statistical analysis to find significant difference

Response 15: As the reviewer suggested the error bars have been added. Statistical analysis requires analyzing large amounts of data to identify common patterns and trends in order to convert them into meaningful information.

The following phrase was inserted in the corrected manuscript: “C2P2.10 sample has a swelling degree with 8% higher than C2P2.10.Ag1 sample and with 15% higher than the C2P2.10.Ag1-IBF sample. Also, the C2P4.10 shows a degree of swelling with 8% higher than the C2P4.10.Ag1 sample and with 6% higher than C2P4.10.Ag1-IBF. There is a greater difference in the swelling degree between C2P2 samples than C2P4 samples.”

Point 16: Discussion

Compare the obtained results with more current study.

Response 16: The obtained results were compared with more current literature studies, as suggested.

Point 17: Discussion must be more elaborative. It would be better to discuss every result and compare with current studies

Response 17: More elaborate discussions were added and the results were compared with data from the literature

Point 18: Conclusion

Conclusion is well justified.

Response 18: Thanks for your appreciation and suggestions

Reviewer 2 Report

The manuscript “Silver nanoparticles biocomposite films with antimicrobial activity: in vitro and in vivo tests” describe the formation of a biocomposite film with antimicrobial properties based on chitosan, poly (vinyl alcohol) and silver nanoparticles. The films were tested at in vitro level on HDFa cell lines and in vivo on mouse model. The authors characterized the formed films by FTIR, SEM, EDX, swelling and release capacity. The paper has sufficient scientific data and evidence to prove the designed methodology and application. However, I have few minor comments:

It would be interesting to add here some characterization of prepared nanoparticles: size, shape and conc. chosen to incorporate in the films.

Figure 3: Do nanoparticles maintain their structure after embedding into the membrane. Is it possible to see them under TEM onto the membrane surface?

Why authors have only chosen S. aureus to test the antibacterial activity. It would be nice to include some gram-negative strains as well.

Author Response

Response to Reviewer 2 Comments

The manuscript “Silver nanoparticles biocomposite films with antimicrobial activity: in vitro and in vivo tests” describe the formation of a biocomposite film with antimicrobial properties based on chitosan, poly (vinyl alcohol) and silver nanoparticles. The films were tested at in vitro level on HDFa cell lines and in vivo on mouse model. The authors characterized the formed films by FTIR, SEM, EDX, swelling and release capacity. The paper has sufficient scientific data and evidence to prove the designed methodology and application. However, I have few minor comments:

Point 1: It would be interesting to add here some characterization of prepared nanoparticles: size, shape and conc. chosen to incorporate in the films.

Response 1: Both the preparation method and the characterization methods for silver nanoparticles were added to the manuscript

Point 2: Figure 3: Do nanoparticles maintain their structure after embedding into the membrane. Is it possible to see them under TEM onto the membrane surface?

Response 2: Unfortunately, we do not have the possibility to do cryo-TEM analysis. To highlight the presence of AgNps, Figure S1 was added in the supplementary material with the results of UV-Vis spectroscopy.

Point 3: Why authors have only chosen S. aureus to test the antibacterial activity. It would be nice to include some gram-negative strains as well.

Response 3: The results of the antimicrobial tests on 2 gram-negative strains were also included in the manuscript, as recommended by the reviewer.

Round 2

Reviewer 1 Report

The revised version is good but check all references in the text is correctly cited.